# Spot the Key, Recover the Rest: Dual-Path&View Representation Learning for Text-Video Retrieval

## Abstract

In recent years, CLIP-based text-video retrieval methods have developed rapidly, with research primarily focusing on exploiting diverse textual and visual cues to achieve effective feature interaction. However, an accurate retrieval model not only requires strong feature enhancement techniques, *e.g.*, text expansion, but also needs coarse-fine granularity interaction strategies, *e.g.*, word-patch. To overcome the limitations of these two types of challenges, we propose a novel text-video retrieval framework, **SKRR**, *i.e.*, **S**pot the **K**ey, **R**ecover the **R**est, which consists of the Dual-Path Feature Partitioning module (DPFP) for feature enhancement and the Dual-View Feature Interaction module (DVFI) for feature interaction. For DPFP, we simulate the human macro-level cognitive perspective by partitioning visual features into two categories based on their relevance to the text query, and supplementing the less relevant features with additional textual. For DVIF, we simulates the human alignment strategy from macro- to micro-level, effectively focusing on local visual features and comprehensively considering fine-grained interactions. DPFP and DVFI collaborate synergistically, jointly promoting cross-modal feature enhancement and interaction. We evaluate SKRR model on five benchmark datasets, including MSRVTT (50.5 %on R@1), achieving state-of-the-art retrieval performance. Code will be released soon.

## 1 Introduction

With the rapid development of the Internet, massive amounts of unlabeled video data are continuously uploaded and shared. The goal of text-video retrieval is to identify videos from massive unlabeled collections that match a given textual query. Recently, the large-scale text-image pre-trained model CLIP Radford et al. (2021) has achieved remarkable success in various multimodal tasks, *e.g.*, visual question answering, classification, and retrieval, demonstrating strong vision-language alignment capabilities. Existing retrieval methods Luo et al. (2022); Gorti et al. (2022) typically leverage CLIP to project both text and video into a shared latent space, thereby establishing feature level similarity relationship. Meanwhile, to establish this reliable matching relationship, a variety of feature enhancement methods Wu et al. (2023); Wang et al. (2024) and feature interaction strategies Wang et al. (2023); Ma et al. (2024) have been proposed. For example, X-Pool Gorti et al. (2022) employs an attention mechanism to extract the most relevant video frames for a given text, thereby enhancing the representational capacity of visual. UCoFiA Wang et al. (2023) proposes a unified coarse-to-fine alignment model that improves text-video retrieval performance from a comprehensive perspective.

Although existing methods perform well in retrieval tasks, they still fall short when dealing with sparse text queries and complex visual cues. We illustrate this problem with the query *"Query1527: pekids using a mobile watch"* and several paired video frames, as shown in Figure 1(a). Among these frames, only those marked with green boxes are relevant to the query, while the remaining frames marked with red boxes differ from it. When simulating human cognition, we tend to focus on visual cues that match the text and repeatedly compare the keywords in the query with the visual entities in the frames, such as *"pekids"* and *"watch."* In other words, this iterative comparison process enables the determination of the optimal cross-modal feature interaction. Moreover, the ignored visual cues can be represented using additional text, *e.g.*, *"a little girl in red is carefully reading about the watch"*. However, this human-aligned approach is rarely considered in existing methods Fang et al.

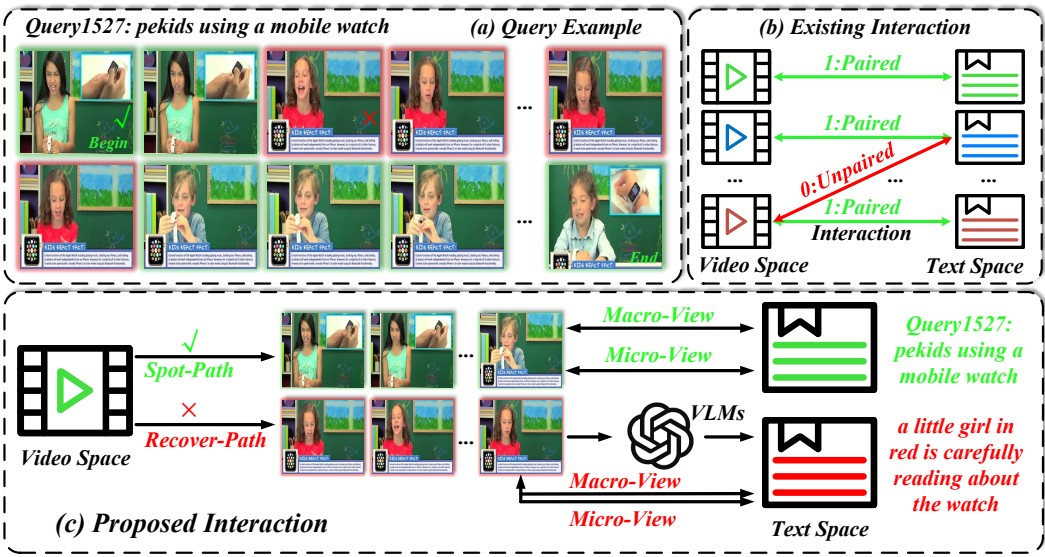

Figure 1: **Motivation.** (a) A query example showing that not all visual features are relevant to the text query. Green boxes indicate relevant frames, while red boxes indicate irrelevant ones. (b, c) Comparison between existing methods and our proposed text-video feature interaction strategy.

(2023); Wang et al. (2024); Hur et al. (2025), which often indiscriminately align all visual features with the text, inevitably hindering subsequent feature alignment, as shown in Figure 1(b).

Based on those analysis, we summarize feature enhancement as consisting of two paths and feature interaction as consisting of two views, as shown in Figure 1(c). The two paths are: **(1)** *Spot-Path, i.e., "Spot the Key"*, which focuses on the content deemed important; **(2)** *Recover-Path, i.e., "Recover the Rest"*, which supplements the content considered missing. The two views are: **(1)** *Macro-View, i.e.,* the interaction between video frames and text sentences at a global level, representing alignment of the overall features; **(2)** *Micro-View, i.e.,* the interaction between visual entities and keywords at a local level, representing alignment of fine-grained features. The two path strategies not only align with the selective mechanism in human macro-level cognition but also ensure the accuracy of subsequent feature interactions. The two-view strategy not only aligns with the human cognitive process from macro to micro levels but also facilitates interactions across features of different granularities.

Based on the above summary, we propose a novel text-video retrieval framework, **SKRR**, *i.e.,* **S**pot the **K**ey, **R**ecover the **R**est. Figure 2 illustrates the over all framework in this paper. **First**, we propose a **D**ual-**P**ath **F**eature **P**artitioning module (**DPFP**), which comprises a Spot-Path that selects high-similarity frames based on global sentence-frame similarity for fully informative interactions, and a Recover-Path that leverages VLMs, *e.g.*, GPT-4, to generate textual descriptions for low-similarity frames as complementary interaction targets. **Second**, we propose a **D**ual-**V**iew **F**eature **I**nteraction module (**DVFI**) to further illustrate the interaction between text and video features from a perspective consistent with human cognition. At the macro level, DVFI aligns video frames with sentence features, while at the micro level, it aligns video patches with word features. To address patch redundancy, we aggregate and re-represent patch features with attention, enabling fine-grained alignment without being constrained by trivial details or background. **Third**, DPFP and DVFI collaborate synergistically, jointly promoting cross-modal feature enhancement and interaction.

We summarize our contributions as follows: (1) We propose a novel text-video retrieval framework, SKRR, which integrates DPFP for feature enhancement and DVFI for feature interaction. (2) The proposed DPFP simulates the human macro-level perspective by partitioning visual features and supplementing textual ones, thereby enhancing cross-modal interaction accuracy. (3) The proposed DVFI simulates the human alignment strategy from macro- to micro-level, effectively focusing on local visual features and comprehensively considering fine-grained interactions. (4) We conduct

extensive experiments on five benchmark datasets, including MSRVTT, DiDeMo, LSMDC, ActivityNet, and Charades, and achieve state-of-the-art retrieval performance (50.5% R@1 on MSRVTT).

## 2 RELATED WORK

**Text-Video Retrieval.** Text-video retrieval aims to retrieve the most semantically relevant video from a large collection based on a given textual query. Early works Liu et al. (2019); Gabeur et al. (2020) primarily focus on enhancing feature representations to align text and video, as well as on establishing benchmarks and foundational models. With the large-scale text-image pretraining model CLIP Radford et al. (2021) achieving significant success, it has inspired improvements in retrieval tasks. For example, CLIP4Clip Luo et al. (2022) transfers the knowledge of the CLIP model to video-language retrieval in an end-to-end manner. Recently, transformer-based text-video retrieval methods Gorti et al. (2022); Liu et al. (2022) use cross-attention to abstract multimodal cues, achieving significant performance gains. TS2Net Liu et al. (2022) employs a "token shift and selection transformer" to preserve token integrity and capture subtle actions, improving retrieval performance. HBI Jin et al. (2023a) performs hierarchical representation by clustering frame-level features and establishes a game-theoretic fine-grained alignment, which has attracted considerable attention. Additionally, a large number of studies Wu et al. (2023); Wang et al. (2024); Xiao et al. (2025) focus on enhancing textual features. Cap4Video Wu et al. (2023) leverages LLM for zero-shot video captioning, expanding the semantic scope of textual at the explicit semantic level.

**Feature Enhancement.** For **video** feature enhancement, most methods Lei et al. (2021); Luo et al. (2022); Gorti et al. (2022) employ transformers to aggregate features. For example, X-Pool Gorti et al. (2022) employs an attention mechanism to extract the most relevant video frames corresponding to a given text, thereby enhancing the representation capability of visual features. For **text** feature enhancement, existing methods can be categorized into implicit Wang et al. (2024); Xiao et al. (2025) and explicit Wu et al. (2023); Bai et al. (2025) approaches. For example, T-Mass Wang et al. (2024) enhances the representation of textual features at the implicit semantic level through random text modeling and regularization techniques. GQE Bai et al. (2025) proposes a unified text enrichment framework that leverages VLMs for event-level captioning and LLMs for query diversification, thereby improving retrieval performance at the explicit level.

**Feature Interaction.** Feature interaction refers to the process of aligning cross-modal text and video features. Existing works Jin et al. (2023a;c) mainly focus on coarse-to-fine and deep-level alignments. For example, UCoFiA Wang et al. (2023) proposes a unified coarse-to-fine alignment model that effectively improves text-video retrieval performance from a comprehensive perspective. EERCF Tian et al. (2024a) adopts multi-granularity visual feature learning to ensure that the model captures visual content from abstract to detailed levels during training. Additionally, UATVR Fang et al. (2023) proposes an uncertainty-adaptive retrieval framework that models features with probabilistic embedding to mitigate hierarchical discrepancies and enhance retrieval performance.

Feature enhancement and interaction are the main approaches for text-video retrieval. However, due to the inherent modality gap between text and video, existing methods often adopt brute-force strategies, which implicitly assume that the features from both modalities are fully equivalent. Feature enhancement is non-specific, relying on automatic selection, while feature interaction is incomplete due to unbalanced cross-modal features. Therefore, our work focuses on efficient feature enhancement and comprehensive feature interaction to improve retrieval performance.

## 3 METHODOLOGY

### 3.1 FEATURE EXTRACTION AND INTERACTION

**Feature Extraction.** Let $\mathcal{D} = (\mathcal{T}, \mathcal{V})$ denote a language and vision dataset, where $\mathcal{T}$ is a set of texts, and $\mathcal{V}$ is a set of videos. The goal of text-video retrieval is to rank the relevance between a text query $t \in \mathcal{T}$ and video set $\mathcal{V}$. Recent works Luo et al. (2022); Jin et al. (2023a); Wang et al. (2024) have shown CLIP's Radford et al. (2021) strong performance in modality feature representation, inspiring us to employ CLIP as our backbone. Specifically, a video $v \in \mathcal{V}$ consists of $N_f$ sequential frames $[f_1, f_2, ..., f_{N_f}] \in \mathbb{R}^{N_f \times H \times W \times C}$, where each frame is divided into $N_p$ patches $[p_1, p_2, ..., p_{N_p}] \in \mathbb{R}^{N_p \times P \times P \times C}$ with $P \times P$ size. Following Radford et al. (2021); Luo et al. (2022), we utilize the

CLIP visual encoder to extract the patch features $V_p = [p_1, \ldots, p_{N_p}] \in \mathbb{R}^{N_p \times D}$ for each frame, and set $p_0$ as the [CLS] token of the current frame. We aggregate the [CLS] tokens of all video frames to obtain the frame features $V_f = [f_1, f_2, \ldots, f_{N_f}] \in \mathbb{R}^{N_f \times D}$. Similarly, given a text query $t \in \mathcal{T}$, we leverage the CLIP text encoder to extract the word features $T_w = [w_1, w_2, \ldots, w_{N_w}] \in \mathbb{R}^{N_w \times D}$, where $N_w$ denotes the length of the word sequence, and set the [BOS] and [EOS] tokens as the beginning and end of the sequence, respectively. We take the representation of the [EOS] token as the sentence feature $T_s = [s] \in \mathbb{R}^{1 \times D}$. In summary, by leveraging the pretrained feature extractors, we obtain the video features $V_f$ and $V_p$, text features $T_s$ and $T_w$, as shown in Figure. 2(a).

**Feature Interaction.** Feature interaction refers to the process of computing the similarity between cross-modal features. Here, we take fine-grained word features $T_w \in \mathbb{R}^{N_w \times D}$ and patch features $V_p \in \mathbb{R}^{N_p \times D}$ as an example to illustrate this process. First, the alignment matrix is defined as $A = [a_{ij}] \in \mathbb{R}^{N_w \times N_p}$, where $a_{ij} = \frac{w_i \cdot p_j}{||w_i|| \cdot ||p_j||}$ represents the alignment score between the $i_{th}$ word feature and the $j_{th}$ patch feature. For the $i_{th}$ word feature, we calculate its maximum alignment score as $\max_j a_{ij}$, and use the weighted average maximum alignment score over all word features as the word-to-patch similarity. Similarly, we can also obtain the patch-to-word maximum alignment score $\max_i a_{ij}$, and the total word-patch similarity score can be defined as:

$$S_{T_w, V_p} = \frac{1}{2} \Big( \sum_{i=1}^{N_w} \theta_w^i \max_j a_{ij} + \sum_{j=1}^{N_p} \theta_p^j \max_i a_{ij} \Big), \tag{1}$$

where $\theta_w = [\theta_w^1, \theta_w^2, \ldots, \theta_w^{N_w}] = \text{Softmax}(\text{MLP}_w(T_w))$ and $\theta_p = [\theta_p^1, \theta_p^2, \ldots, \theta_p^{N_p}] = \text{Softmax}(\text{MLP}_p(V_p))$ are the weights of the word features and patch features, respectively. If the number of features on one side is 1, $e.g.$, the sentence feature $T_s = [s] \in \mathbb{R}^{1 \times D}$, the sentence-patch similarity score is defined as $S_{T_s, V_p} = \sum_{i=1}^{N_p} \theta_p^i a_i$, and $a_i = \frac{s \cdot p_i}{||s|| \cdot ||p_i||}$. Finally, the word-patch cross-modal contrastive loss can be formulated as:

$$\mathcal{L}_{T_w, V_p} = -\frac{1}{2} \frac{1}{B} \sum_{i=1}^{B} \Big( \log \frac{\exp(S_{T_w^i, V_p^i}/\tau)}{\sum_{j=1}^{B} \exp(S_{T_w^i, V_p^j}/\tau)} + \log \frac{\exp(S_{T_w^i, V_p^i}/\tau)}{\sum_{j=1}^{B} \exp(S_{T_w^j, V_p^i}/\tau)} \Big), \tag{2}$$

where $B$ is the batch size and $\tau$ is the temperature hyper-parameter. This loss function maximizes the similarity of positive pairs and minimizes the similarity of negative pairs. Similarly, $\mathcal{L}_{T_w, V_f}$, $\mathcal{L}_{T_s, V_p}$, and $\mathcal{L}_{T_s, V_f}$ can also be computed.

## 3.2 DUAL-PATH FEATURE PARTITIONING

Although the CLIP visual encoder can extract powerful frame and patch features from videos, not all of them are strongly correlated with the text. In other words, to achieve more effective feature interactions, both sides of the interaction need to provide informative representations. Therefore, in this section, we propose a **D**ual-**P**ath **F**eature **P**artitioning module (**DPFP**), which consists of the Spot-Path that provides fully informative interaction targets and the Recover-Path that provides complementary interaction targets, as illustrated in Figure 2 (b).

**Spot the Key: Spot-Path.** As the Du Fu once said, "Reaching the summit, all other mountains appear small." We believe that grasping the interaction targets from a global and holistic perspective is essential. First, when aligning text and video, we, as humans, first read the overall sentence feature $T_s = [s] \in \mathbb{R}^{1 \times D}$, and then compare it with each video frame features $V_f = [f_1, f_2, \ldots, f_{N_f}] \in \mathbb{R}^{N_f \times D}$ to identify the effective frames $V_f^+ \in \mathbb{R}^{N_f^+ \times D}$, where $N_f^+ < N_f$. Next, from a human-centric perspective, the selection of effective frames $V_f^+$ can be directly based on the similarity $S_{T_s, V_f}$ between $T_s$ and $V_f$. Finally, we select the *top-$N_f^+$* frames $V_f^+$ from $V_f$ based on $S_{T_s, V_f}$:

$$V_f^+ = \text{Top}N_f^+(V_f, S_{T_s, V_f}), \quad S_{T_s, V_f} = \frac{T_s \cdot V_f}{||T_s|| \cdot ||V_f||} \in \mathbb{R}^{1 \times N_f} \tag{3}$$

In addition, the patch features are also updated from $V_p \in \mathbb{R}^{(N_f \cdot N_p) \times D}$ to $V_p^+ \in \mathbb{R}^{(N_f^+ \cdot N_p) \times D}$.

**Recover the Rest: Recover-Path.** As Feynman once said, *"What I cannot create, I do not understand."* We believe that the missing interaction objects should be recovered to reduce information loss. First, the lost visual features, including $V_f^- = V_f - V_f^+$ and $V_p^- = V_p - V_p^+$, exhibit low similarity with the text. They should be preserved reasonably rather than discarded directly, to avoid excessive loss of visual information. By leveraging existing text augmentation methods Wu et al. (2023); Ma et al. (2024), we employ large visual-language models, *e.g.*, GPT-4 OpenAI (2023), to

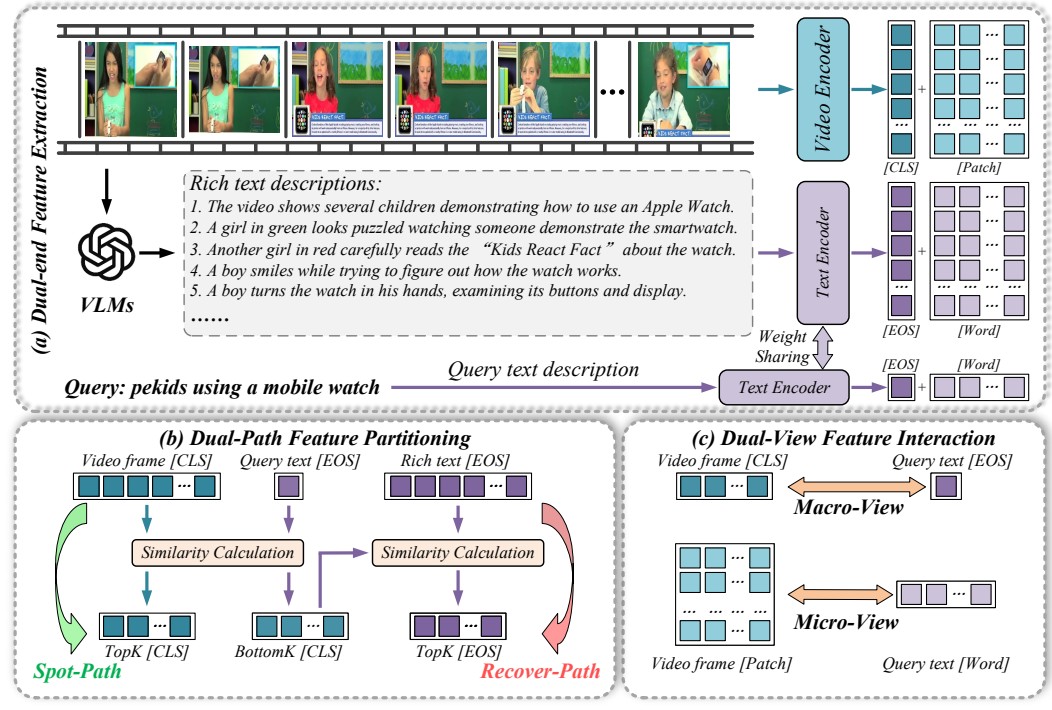

Figure 2: **Framework.** (a) Dual-end Feature Extraction is used to extract features from both text and video. (b) Dual-Path Feature Partitioning selects frames most relevant to the query based on similarity and provides textual descriptions for the less relevant frames. (c) Dual-View Feature Interaction reconsiders feature interactions at both the macro and micro levels.

generate textual descriptions of videos in a zero-shot manner, and obtain the corresponding textual features $T_s^= = [s_1, s_2, ..., s_{N_s^=}] \in \mathbb{R}^{N_s^= \times D}$. Second, to obtain the text that is strongly paired with $V_f^-$ and $V_p^-$, we follow the idea of the Spot-Path to generate the augmented text $T_s^- \in \mathbb{R}^{N_s^- \times D}$:

$$T_s^- = \text{Top}N_s^-(T_s^=, S_{T_s^=, V_f^-}), \quad S_{T_s^=, V_f^-} = \frac{T_s^= \cdot V_f^-}{||T_s^=|| \cdot ||V_f^-||} \in \mathbb{R}^{N_s^= \times 1}, \tag{4}$$

where $V_f^- = \frac{1}{N_f^-} \sum_{i=1}^{N_f^-} V_{f,i}^-$. Similarly, the word features are also updated from $T_w^= \in \mathbb{R}^{(N_s^= \cdot N_w) \times D}$ to $T_w^- \in \mathbb{R}^{(N_s^- \cdot N_w) \times D}$. In summary, under the Spot-Path and Recover-Path, we obtain enhanced textual features ($T_s^+ = T_s, T_w^+ = T_w$ and $T_s^-, T_w^-$) and enhanced video features ($V_f^+, V_p^+$ and $V_f^-, V_p^-$) that are more representative than the original ones.

## 3.3 DUAL-VIEW FEATURE INTERACTION

In Section 3.2, we obtain both coarse-grained features ($T_s^+, T_s^-, V_f^+, V_f^-$) and fine-grained features ($T_w^+, T_w^-, V_p^+, V_p^-$) that are ready for direct interaction. While coarse-grained features allow direct interaction, large and redundant fine-grained features such as $V_p^+$ require additional processing. Therefore, in this section, we propose a **D**ual-**V**iew **F**eature **I**nteraction module (**DVFI**) to model interactions between text and video features in a manner consistent with human cognition.

**Macro Feature Interaction: Macro-View.** Macro-view feature interaction refers to coarse-grained feature interaction. Specifically, we directly compute the similarity between sentence features and frame features based on Equation 1:

$$S_{T_s^+, V_f^+} = \sum_{i=1}^{N_f^+} \theta_f^i a_i, \quad S_{T_s^-, V_f^-} = \sum_{i=1}^{N_f^-} \theta_f^i a_i. \tag{5}$$

Compared with $S_{T_s,V_f}$, Equation 5 not only partitions visual features from a macro perspective but also supplements textual information, thereby providing a clear advantage in the accuracy of feature interaction. Finally, the interaction losses $\mathcal{L}_{T_s^+,V_f^+}$ and $\mathcal{L}_{T_s^-,V_f^-}$ are computed using Equation 2.

**Micro Feature Interaction: Micro-View.** The features involved in Macro-View interaction exhibit a high degree of simplicity in both quantity and representation. However, high-volume and redundant patch features cannot participate in interaction with text in the same way. For example, if we set $N_f^+ = N_f^- = \frac{1}{2}N_f$, the number of patch features obtained is $\frac{1}{2}N_f \times \frac{H \times W \times C}{P \times P \times C} = 294$ for $N_f = 12$ and *CLIP-ViT-B/32* (1176 for *CLIP-ViT-B/16*). To reduce the number of these features while enhancing their representativeness, previous methods Liu et al. (2022); Wang et al. (2023) primarily decrease the number of patch features by selecting those with higher scores. It is worth noting that these methods often lack targeted feature selection, which may lead to information loss. In summary, patch features involved in interaction with text should be region-specific rather than isolated.

To reduce the number of patch features and obtain important merged patch regions, we propose a **P**atch **F**eatures **C**ompression **M**odule (**PFCM**) from a human micro-view, as illustrated in Figure 2(c). First, we focus on the Spot-Path patch features $V_p^+ = N_f^+ \times [p_1, p_2, \ldots, p_{N_p}] \in \mathbb{R}^{N_f^+ \times N_p \times D}$. This step, benefiting from global features selection, not only focuses on key frames but also significantly reduces the number of patch features. Next, we further compress the patch features $V_p^+$ from a human cognitive perspective. Specifically, we utilize a variant of the $k$-nearest neighbor-based density peaks clustering algorithm (DPC-KNN) Du et al. (2016). Given patch features $V_p^+$, we compute the local density $\rho_i$ of each patch $p_i$ according to its $k$-neatest neighbors $\rho_i = \exp(-\frac{1}{k}\sum \|p_i - p_j\|_2)$, where $p_j \in \mathrm{KNN}(p_i)$ denotes the $k$-neatest neighbors of the patch $p_i$. Then, we compute the distance indicator $\delta_i$ of each patch:

$$\delta_i = \begin{cases} \min \|p_i - p_j\|_2, & \rho_i < \rho_j, \\ \max \|p_i - p_j\|_2, & \rho_i \geq \rho_j. \end{cases} \tag{6}$$

Intuitively, the patch $p_i$ with a larger local density $\rho_i$ and distance indicator $\delta_i$ is more likely to become a cluster center. We determine a cluster center by selecting the patches with the highest scores $\rho_i \times \delta_i$, and then merge the neighboring patches. To ensure that important patches contribute more to the output and capture long-range dependencies, we introduce an importance score $I$ and an attention mechanism. Given the cluster center $p_i$ and the corresponding $i^{th}$ cluster $C_i$, the merged patch $p_i^* = \frac{\sum_{j \in C_i} \exp(I_j) p_j}{\sum_{j \in C_i} \exp(I_j)}$, where $I_j = \mathrm{MLP}_p(p_i)$. The merged patch $p_i^*$ is fed into a transformer block as query $Q$, and the original patch $p_i$ is used as key $K$ and value $V$, and the importance score $I$ is added to the attention weight as follows:

$$\mathrm{Attention}(Q, K, V) = \mathrm{softmax}\left(QK^T/\sqrt{d_k} + I\right)V, \tag{7}$$

where $d_k$ represents the feature dimension. By introducing the patch importance score $I$ and the attention mechanism, PFCM not only reduces the number of patch features but also focuses on key features and spatial relationships. Finally, we iteratively apply PFCM to compress and aggregate patch features, reducing their number. Similar to the operations in Equation 1, we can compute the similarity between text word features and aggregated patch features:

$$S_{T_w^+,V_p^+} = \frac{1}{2}(\sum_{i=1}^{N_w^+} \theta_w^i \max_j a_{ij} + \sum_{j=1}^{N_p^+} \theta_p^j \max_i a_{ij}),$$

$$S_{T_w^-,V_p^-} = \frac{1}{2}(\sum_{i=1}^{N_w^-} \theta_w^i \max_j a_{ij} + \sum_{j=1}^{N_p^-} \theta_p^j \max_i a_{ij}). \tag{8}$$

Furthermore, the sharp reduction in the number of patch features enhances their granularity, enabling them to participate in interaction with sentence feature:

$$S_{T_s^+,V_p^+} = \sum_{i=1}^{N_p^+} \theta_p^i a_i, \quad S_{T_s^-,V_p^-} = \sum_{i=1}^{N_p^-} \theta_p^i a_i. \tag{9}$$

In the granularity-based interactions, we omit $S_{T_w,V_f}$: the granularity gap is too large to align with human cognition, and experiments show its performance is suboptimal. Consequently, the interaction losses $\mathcal{L}_{T_w^+,V_p^+}$, $\mathcal{L}_{T_w^-,V_p^-}$, $\mathcal{L}_{T_s^+,V_p^+}$, and $\mathcal{L}_{T_s^-,V_p^-}$ are computed using Equation 2.

## 3.4 TRAINING AND SAMPLING

**Training.** Under the Spot-Path and Recover-Path, we can obtain all cross-modal contrastive losses:

$$\mathcal{L}_+ = \mathcal{L}_{T_s^+,V_f^+} + \mathcal{L}_{T_s^+,V_p^+} + \mathcal{L}_{T_w^+,V_p^+}, \quad \mathcal{L}_- = \mathcal{L}_{T_s^-,V_f^-} + \mathcal{L}_{T_s^-,V_p^-} + \mathcal{L}_{T_w^-,V_p^-}. \tag{10}$$

| Methods | MSRVTT (Text-to-Video) | | | | | MSRVTT (Video-to-Text) | | | | |
|---|---|---|---|---|---|---|---|---|---|---|
| | R@1↑ | R@5↑ | R@10↑ | MdR↓ | MnR↓ | R@1↑ | R@5↑ | R@10↑ | MdR↓ | MnR↓ |
| *CLIP-ViT-B/32* | | | | | | | | | | |
| Clip4clip Luo et al. (2022) | 44.5 | 71.4 | 81.6 | 2.0 | 15.3 | 42.7 | 70.9 | 80.6 | 2.0 | 11.6 |
| X-Pool Gorti et al. (2022) | 46.9 | 72.8 | 82.2 | 2.0 | 14.3 | 44.4 | 73.3 | 84.0 | 2.0 | 9.0 |
| UATVR Fang et al. (2023) | 47.5 | 73.9 | 83.5 | 2.0 | 12.3 | 46.9 | 73.8 | 83.8 | 2.0 | 8.6 |
| HBI Jin et al. (2023a) | 48.6 | 74.6 | 83.4 | 2.0 | 12.0 | 46.8 | 74.3 | 84.3 | 2.0 | 8.9 |
| Cap4Video Wu et al. (2023) | 49.3 | 74.3 | 83.8 | 2.0 | 12.0 | 47.1 | 73.7 | 84.3 | 2.0 | 8.7 |
| TeachCLIP Tian et al. (2024b) | 46.8 | 74.9 | 82.9 | - | - | - | - | - | - | - |
| Mv-adapter Jin et al. (2024) | 46.2 | 73.2 | 82.7 | - | - | 47.2 | 74.8 | 83.9 | - | - |
| T-Mass Wang et al. (2024) | 50.2 | 75.3 | 85.1 | 1.0 | 11.9 | 47.7 | **78.0** | 86.3 | 2.0 | 8.0 |
| EERCF Tian et al. (2024a) | 47.8 | 74.1 | 84.1 | - | - | 44.7 | 74.2 | 83.9 | - | - |
| DiscoVLA Shen et al. (2025) | 47.0 | 73.0 | 82.8 | - | 14.1 | 47.7 | 73.6 | 83.6 | - | 10.0 |
| BiHSSP Liu et al. (2025) | 48.1 | 74.0 | 84.1 | 2.0 | 12.1 | 48.0 | 74.1 | 83.5 | 2.0 | 9.0 |
| SKRR | **50.5** | **76.1** | **86.2** | **1.0** | **11.3** | **48.3** | 76.2 | **86.7** | **2.0** | **7.8** |
| *CLIP-ViT-B/16* | | | | | | | | | | |
| X-Pool Gorti et al. (2022) | 48.2 | 73.7 | 82.6 | 2.0 | 12.7 | 46.4 | 73.9 | 84.1 | 2.0 | 8.4 |
| UATVR Fang et al. (2023) | 50.8 | 76.3 | 85.5 | 1.0 | 12.4 | 48.1 | 76.3 | 85.4 | 2.0 | 8.0 |
| Cap4Video Wu et al. (2023) | 51.4 | 75.7 | 83.9 | 1.0 | 12.4 | 49.0 | 75.2 | 85.0 | 2.0 | 8.0 |
| T-Mass Wang et al. (2024) | 52.7 | 77.1 | 85.6 | 1.0 | 10.5 | 50.9 | **80.2** | **88.0** | 1.0 | 7.4 |
| EERCF Tian et al. (2024a) | **54.1** | 78.8 | 86.9 | - | - | 51.0 | 77.8 | 85.7 | - | - |
| DiscoVLA Shen et al. (2025) | 50.5 | 75.6 | 83.8 | - | 12.1 | 49.2 | 76.0 | 84.7 | - | 8.6 |
| BiHSSP Liu et al. (2025) | 50.8 | 75.9 | 84.4 | 1.0 | 11.0 | 50.3 | 75.5 | 84.5 | 1.5 | 7.8 |
| SKRR | 53.2 | **78.9** | **87.2** | **1.0** | **10.3** | **51.2** | 78.9 | 86.5 | **1.0** | **7.4** |

Table 1: Text-to-video and video-to-text retrieval performance on the MSRVTT Xu et al. (2016).

To prevent data leakage during the sampling process, *i.e.*, avoiding the use of text to partition video frames, we also consider the interaction between the original text and video features:

$$\mathcal{L} = \mathcal{L}_{T_s,V_f} + \mathcal{L}_{T_s,V_p} + \mathcal{L}_{T_w,V_p}. \tag{11}$$

To further improve sampling performance, we optimize the KL divergence over the similarities:

$$\mathcal{L}_+^{KL} = \mathbb{E}_{T,V}[\mathrm{KL}(S_{T_s,V_f}||S_{T_s^+,V_f^+}) + \mathrm{KL}(S_{T_s,V_p}||S_{T_s^+,V_p^+}) + \mathrm{KL}(S_{T_w,V_p}||S_{T_w,V_p^+})],$$

$$\mathcal{L}_-^{KL} = \mathbb{E}_{T,V}[\mathrm{KL}(S_{T_s,V_f}||S_{T_s^-,V_f^-}) + \mathrm{KL}(S_{T_s,V_p}||S_{T_s^-,V_p^-}) + \mathrm{KL}(S_{T_w,V_p}||S_{T_w,V_p^-})]. \tag{12}$$

We aggregate all similarity scores to compute the total similarity loss $\mathcal{L}_{total}$:

$$\mathcal{L}_{total} = \mathcal{L} + \mathcal{L}_+ + \mathcal{L}_- + \mathcal{L}_+^{KL} + \mathcal{L}_-^{KL}. \tag{13}$$

**Sampling.** After training, we compute all similarities, including $S_{T_s,V_f}$, $S_{T_s,V_p}$, and $S_{T_w,V_p}$, which are then aggregated into a final similarity matrix to calculate the corresponding retrieval metrics.

## 4 EXPERIMENTS

### 4.1 EXPERIMENTAL SETTINGS

We adopt five benchmark datasets for the evaluation, including MSRVTT Xu et al. (2016), DiDeMo Anne Hendricks et al. (2017), LSMDC Rohrbach et al. (2015), ActivityNet Krishna et al. (2017) and Charades Sigurdsson et al. (2016). We evaluate retrieval performance using Recall at rank K (R@K, K=1,5,10), Median Rank (MdR), and Mean Rank (MnR). Higher R@K values, together with lower MdR and MnR values, indicate better retrieval performance. The model backbone is initialized from pre-trained CLIP-ViT-B/32. More experimental settings are provided in Appendix A.1.

### 4.2 COMPARISON WITH STATE-OF-THE-ART

We compare the performance of SKRR with recent state-of-the-art methods and list the results in Table 1 and Table 2. SKRR consistently achieves leading retrieval performance across all five datasets. On MSRVTT, SKRR based on CLIP-ViT-B/32 improves the text-to-video retrieval metric R@1 from 50.2 to 50.5 **(+0.6%)** compared to the text expansion model T-Mass Wang et al. (2024), while based on ViT-B/16, it improves R@10 from 85.6 to 87.2 **(+1.9%)**. Meanwhile, SKRR continues to demonstrate strong performance in long-video retrieval tasks such as DiDeMo Anne Hendricks et al.

| Methods | DiDeMo (Text-to-Video) | | | | | LSMDC (Text-to-Video) | | | | |
|---|---|---|---|---|---|---|---|---|---|---|
| | R@1↑ | R@5↑ | R@10↑ | MdR↓ | MnR↓ | R@1↑ | R@5↑ | R@10↑ | MdR↓ | MnR↓ |
| CE Liu et al. (2019) | 16.1 | 41.1 | - | 8.3 | 43.7 | 11.2 | 26.9 | 34.8 | 25.3 | - |
| EMCL-Net Jin et al. (2022) | 45.3 | 74.2 | 82.3 | 2.0 | 12.3 | 23.9 | 46.4 | 53.7 | 8.0 | - |
| TS2-Net Liu et al. (2022) | 41.8 | 71.6 | 82.0 | 2.0 | 14.8 | 23.4 | 42.3 | 50.9 | 9.0 | 56.9 |
| X-Pool Gorti et al. (2022) | 44.6 | 73.2 | 82.0 | 2.0 | 15.4 | 25.2 | 43.7 | 53.5 | 8.0 | 53.2 |
| CLIP-VIP Xue et al. (2022) | 48.6 | **77.1** | 84.4 | 2.0 | - | 25.6 | 45.3 | 54.4 | 8.0 | - |
| DiCoSA Jin et al. (2023b) | 45.7 | 74.6 | 83.5 | 2.0 | 14.7 | 25.4 | 43.6 | 54.0 | 8.0 | 41.9 |
| DiffusionRet Jin et al. (2023c) | 46.7 | 74.7 | 82.7 | 2.0 | 14.3 | 24.4 | 43.1 | 54.3 | 8.0 | 40.7 |
| UATVR Fang et al. (2023) | 43.1 | 71.8 | 82.3 | 2.0 | 15.1 | - | - | - | - | - |
| SKRR | **51.0** | 75.3 | **84.7** | **1.0** | **13.6** | **29.3** | **49.6** | **58.4** | **6.0** | **40.3** |

| Methods | ActivityNet (Text-to-Video) | | | | | Charades (Text-to-Video) | | | | |
|---|---|---|---|---|---|---|---|---|---|---|
| | R@1↑ | R@5↑ | R@10↑ | MdR↓ | MnR↓ | R@1↑ | R@5↑ | R@10↑ | MdR↓ | MnR↓ |
| ClipBERT Lei et al. (2021) | 21.3 | 49.0 | 63.5 | 6.0 | - | 6.7 | 17.3 | 25.2 | 32.0 | 149.7 |
| Clip4clip Luo et al. (2022) | 40.5 | 72.4 | 83.6 | 2.0 | 7.5 | 9.9 | 27.1 | 36.8 | 21.0 | 85.4 |
| HBI Jin et al. (2023a) | 42.2 | 73.0 | 84.6 | 2.0 | 6.6 | - | - | - | - | - |
| T-Mass Wang et al. (2024) | - | - | - | - | - | 14.2 | 36.2 | 48.3 | 12.0 | 54.8 |
| SKRR | **47.0** | **76.2** | **86.4** | **2.0** | **6.3** | **19.3** | **42.2** | **53.5** | **8.0** | **49.7** |

Table 2: Text-to-video retrieval performance on the DiDeMo Anne Hendricks et al. (2017), LSMDC Rohrbach et al. (2015), ActivityNet Krishna et al. (2017) and Charades Sigurdsson et al. (2016).

(2017). Compared to the frame-level feature method X-Pool Gorti et al. (2022), the R@1 improves from 44.6 to 51.0 **(+14.3%)**, highlighting the importance of feature selection and compression rather than relying on the model to automatically filter effective frames. Notably, SKRR maintains high retrieval performance even on the sparse text dataset LSMDC Rohrbach et al. (2015). Compared to the external text expansion method CLIP-VIP Xue et al. (2022), SKRR improves R@1 from 25.6 to 29.3 **(+14.5%)**. In summary, SKRR demonstrates outstanding retrieval performance for both sparse text queries and long-video retrieval tasks.

### 4.3 ABLATION STUDY

In Table 3, we evaluate the core modules of SKRR, including DPFP (Spot-Path and Recover-Path) and DVFI (Macro-View and Micro-View). Row 1 represents our baseline, which applies frame feature average pooling as in Luo et al. (2022). Rows 2 and 3 adopt the feature interaction in Equation 1 while partitioning the video frame features. Compared with Row 1, the performance improves from 43.4 to 47.2, demonstrating the effectiveness of the Spot-Path partition and Macro-View interaction. In addition, Row 4 separately illustrates the complementary textual advantage of the Recover-Path, but the introduction of additional text increases the memory cost. Finally, Row 5 combines the DPFP and DVFI modules, achieving 50.5 R@1 retrieval performance.

The proposed DPFP and DVFI modules work synergistically to enhance cross-modal retrieval performance from three perspectives. **(1)** Human cognition. DPFP first selects salient frames and patches ($V_f^+$, $V_p^+$) while discarding irrelevant ones, and DVFI further emphasizes salient objects within these patches, resembling the way humans filter information from global to local cues. **(2)** Alignment accuracy. By filtering unaligned visual content, DPFP improves coarse- and fine-grained alignment, while DVFI alleviates redundancy by iteratively compressing patch features, making alignment with sentence- and word-level text more precise. **(3)** Computational cost. DPFP reduces the number of frames and patches involved ($N_f^+ < N_f$, $N_p^+ < N_p$), and DVFI progressively halves the remaining patches, together substantially lowering feature counts and computational overhead.

In Table 4, we evaluate the performance of interactions at different granularities, consistently incorporating $S_{T_s, V_f} + S_{T_w, V_p}$. By comparing the three groups of results, we find that incorporating $S_{T_s, V_p}$ alone into SKRR model training can improve the overall retrieval performance. This result demonstrates the effectiveness of PCFM in aggregating patch features and the appropriateness of the granularity for cross-granularity alignment.

In Table 5, we compare the effects of different hyperparameters, $N_f^+ = \hbar N_f$ and $N_p^+ = \lambda N_p$, in the DPFP and DVFI modules on retrieval performance and speed, where $\hbar$ and $\lambda$ denote the

| | DPFP$_{SP}$ | DPFP$_{RP}$ | DVFI$_{MaV}$ | DVFI$_{MiV}$ | R@1↑ | R@5↑ | R@10↑ | MnR↓ | TrainTime↓ | TestTime↓ | GPU↓ |
|---|---|---|---|---|---|---|---|---|---|---|---|
| 1 | | | | | 43.4 | 70.9 | 81.1 | 16.4 | 11.5h | 30s | 11.41GB |
| 2 | ✓ | | | | 45.6 | 73.0 | 82.7 | 14.3 | 11.0h | 28s | 11.05GB |
| 3 | ✓ | | ✓ | | 47.2 | 74.5 | 84.4 | 11.6 | 12.3h | 37s | 11.75GB |
| 4 | | ✓ | ✓ | ✓ | 47.9 | 74.4 | 84.7 | 11.8 | 14.2h | 52s | 13.65GB |
| 5 | ✓ | ✓ | ✓ | ✓ | **50.5** | 76.1 | 86.2 | 11.3 | 15.3h | 62s | 14.29GB |

Table 3: Ablation studies of DPFP and DVFI modules on the MSRVTT Xu et al. (2016).

| $S_{T_s,V_f} + S_{T_w,V_p}$ | Spot-Path $(+)$ | | | | Recover-Path $(-)$ | | | | SKRR-Path $(+,-)$ | | | |
|---|---|---|---|---|---|---|---|---|---|---|---|---|
| | R@1↑ | R@5↑ | R@10↑ | MnR↓ | R@1↑ | R@5↑ | R@10↑ | MnR↓ | R@1↑ | R@5↑ | R@10↑ | MnR↓ |
| $+S_{T_s,V_p}$ | 47.2 | 74.5 | 84.4 | 11.6 | 47.9 | 74.4 | 84.7 | 11.8 | **50.5** | 76.1 | 86.2 | 11.3 |
| $+S_{T_w,V_f}$ | 46.3 | 73.4 | 83.8 | 14.1 | 47.4 | 74.2 | 84.1 | 13.8 | 47.6 | 74.5 | 84.2 | 13.5 |
| $+S_{T_s,V_p} + S_{T_w,V_f}$ | 46.8 | 72.6 | 83.9 | 13.5 | 47.5 | 74.8 | 84.2 | 12.5 | 49.2 | 75.3 | 85.4 | 12.6 |

Table 4: Ablation studies on interactions at different granularities on the MSRVTT Xu et al. (2016).

frame partition factor in DPFP and the patch compression factor in PFCM, respectively. In Row 1, DPFP and DVFI are removed as a reference. In the remaining experiments, we find that retrieval performance reaches its optimum when $(\hbar, \lambda) = (0.50, 0.50)$. This result indicates that neither excessive nor overly conservative macro-frame selection and micro-patch compression are desirable; instead, a balance should be achieved at the optimal number of modalities for text-video interaction.

| $\hbar, \lambda$ | $N_f^+$ | $N_p^+$ | R@1↑ | R@5↑ | R@10↑ | MnR↓ | TrainTime↓ | TestTime↓ | GPU↓ |
|---|---|---|---|---|---|---|---|---|---|
| (1.00, 1.00) | 12 | 588 | 44.2 | 71.7 | 82.0 | 15.2 | 13.6h | 98s | 12.43GB |
| (0.75, 0.50) | $12 \xrightarrow{\text{DPFP}} 9$ | $441 \xrightarrow{\text{PFCM}} 221 \xrightarrow{\text{PFCM}} 111 \xrightarrow{\text{PFCM}} 056$ | 45.7 | 72.4 | 83.2 | 14.6 | 15.6h | 61s | 14.78GB |
| (0.50, 0.75) | $12 \xrightarrow{\text{DPFP}} 6$ | $294 \xrightarrow{\text{PFCM}} 187 \xrightarrow{\text{PFCM}} 141 \xrightarrow{\text{PFCM}} 106$ | 48.5 | 75.7 | 85.3 | 12.5 | 16.2h | 68s | 14.56GB |
| (0.50, 0.50) | $12 \xrightarrow{\text{DPFP}} 6$ | $294 \xrightarrow{\text{PFCM}} 147 \xrightarrow{\text{PFCM}} 074 \xrightarrow{\text{PFCM}} 037$ | **50.5** | 76.1 | 86.2 | 11.3 | 15.3h | 62s | 14.29GB |
| (0.50, 0.25) | $12 \xrightarrow{\text{DPFP}} 6$ | $294 \xrightarrow{\text{PFCM}} 074 \xrightarrow{\text{PFCM}} 019 \xrightarrow{\text{PFCM}} 005$ | 49.2 | 76.2 | 85.7 | 12.1 | 14.2h | 49s | 13.78GB |
| (0.25, 0.50) | $12 \xrightarrow{\text{DPFP}} 3$ | $147 \xrightarrow{\text{PFCM}} 074 \xrightarrow{\text{PFCM}} 037 \xrightarrow{\text{PFCM}} 019$ | 48.4 | 75.2 | 85.1 | 12.5 | 14.7h | 55s | 14.02GB |

Table 5: Ablation studies of the DPFP and DVFI modules with hyper-parameters $N_f^+$ and $N_p^+$.

## 4.4 MORE RESULTS

We provide more results in the appendix, including A.1: experiment setups and implementation details; A.2: Video-to-text retrieval performance on the DiDeMo, LSMDC, ActivityNet, and Charades datasets, generalization and video question answering performance; A.3: further ablation studies on DPFP and DVFI modules; A.4: visualization results; A.5: ethics statement.

## 5 CONCLUSION

To overcome the challenges in feature enhancement and cross-granularity interaction for sparse text queries and complex visual cues, we propose a novel text-video retrieval framework, SKRR, *i.e.*, Spot the Key, Recover the Rest. SKRR consists of the Dual-Path Feature Partitioning module (DPFP) for feature enhancement and the Dual-View Feature Interaction module (DVFI) for feature interaction. In DPFP, we simulate the human macro-level cognitive perspective by partitioning visual features into two categories based on their relevance to the text query and supplementing the less relevant features with additional textual information. In DVFI, we simulate the human alignment strategy from macro- to micro-level, effectively focusing on local visual features while comprehensively considering fine-grained interactions. DPFP and DVFI work synergistically, jointly promoting cross-modal feature enhancement and interaction. Experiments on five benchmark datasets demonstrate that the SKRR model achieves state-of-the-art performance, validating the effectiveness of our approach. We hope that this work will provide inspiration to the video retrieval community.

ACKNOWLEDGEMENTS

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

# A APPENDIX

## A.1 EXPERIMENTAL SETTINGS

**Datasets.** We adopt five benchmark datasets for the evaluation, including MSRVTT Xu et al. (2016), DiDeMo Anne Hendricks et al. (2017), LSMDC Rohrbach et al. (2015), ActivityNet Krishna et al. (2017) and Charades Sigurdsson et al. (2016). **MSRVTT** consists of 10,000 YouTube videos, each paired with 20 captions. We follow the training protocol in Yu et al. (2018) and evaluate our model SKRR on both text-to-video and video-to-text retrieval tasks using the 1K-A test split. **DiDeMo** contains 10,464 video clips and 40,543 captions. We concatenate the descriptions of individual video segments to construct a "video-paragraph" for retrieval. We use the training and testing protocols from Gabeur et al. (2020). **LSMDC** includes 118,081 video clips from 202 movies. The duration of videos in the LSMDC dataset is short. We use the split from Torabi et al. (2016), with 1,000 videos reserved for testing. **ActivityNet** contains densely annotated temporal segments for 20,000 YouTube videos. Following Jin et al. (2023a), we report results on the "val1" split, using 10,009 videos for training and 4,917 for testing. **Charades** consists of 9,848 video clips, where each corresponds to a text description. We adopt the split protocol from Lin et al. (2022).

**Metrics.** We evaluate retrieval performance using Recall at rank K (R@K, K=1,5,10, higher is better), Median Rank (MdR, lower is better), and Mean Rank (MnR, lower is better). R@K measures the percentage of test samples whose correct results appear in the top-K retrieved items. MdR reports the median rank of the ground-truth results, and MnR reports their mean rank.

**Implementation Details.** Following previous methods Luo et al. (2022); Gorti et al. (2022), we use CLIP Radford et al. (2021) as the backbone model for both text and video feature extraction. The SKRR model is trained for 5 epochs with a batch size of 32, and the feature dimension $D$ is set to 512. For the MSRVTT, LSMDC, and Charades datasets, we set the frame length to $N_f = 12$ and the word length to $N_w = 32$. For the long video DiDeMo and ActivityNet datasets, we set the frame length to $N_f = 64$ and the word length to $N_w = 64$. We uniformly sample $N_f$ frames from each video clip and resize them to $224 \times 224$ pixels. Accordingly, the number of patches per frame is set to $N_p = \frac{224 \times 224}{32 \times 32} = 49$. We adopt the vision-language model VILA-13B Lin et al. (2024) to generate $N_s^= = 6$ text descriptions for each video. The initial learning rate is set to $1e\text{-}5$ for both the text encoder and the video encoder. We train our model on 4 NVIDIA RTX 4090 24GB GPUs, and the training process takes approximately 15 hours.

## A.2 PERFORMANCE

**Video-to-text Retrieval Performance.** In Table 6, we report the video-to-text retrieval performance on the DiDeMo Anne Hendricks et al. (2017), LSMDC Rohrbach et al. (2015), ActivityNet Krishna et al. (2017) and Charades Sigurdsson et al. (2016) datasets. The video-to-text retrieval task involves finding the matching text given visual features, which is the opposite of the text-to-video retrieval task. Table 6 shows that SKRR consistently improves retrieval performance across these four datasets. For example, on the long-video DiDeMo dataset, SKRR achieves a 1.3-point improvement in R@1 compared to the text-only enhancement method T-Mass Wang et al. (2024), thanks to its visual feature enhancement and text augmentation.

**Cross-domain Generalization Performance.** Cross-domain generalization performance measures the ability of a model to perform on data from unseen domains. In Table 7, we use MSRVTT Xu et al. (2016) as the source domain for training and DiDeMo Anne Hendricks et al. (2017) and LSMDC Rohrbach et al. (2015) as the target domains for testing to evaluate the generalization performance of SKRR. Compared with recent state-of-the-art methods, *e.g.*, T-Mass Wang et al. (2024), SKRR demonstrates consistent performance advantages.

**Video Question Answering Performance.** Let $\mathcal{D} = (\mathcal{Q}, \mathcal{V}, \mathcal{A})$ denote a video question answering dataset, where $\mathcal{Q}$ is a set of questions, $\mathcal{V}$ is a set of videos, and $\mathcal{A}$ is a set of answers. The task of video question answering requires the model to answer a question $q \in \mathcal{Q}$ by referring to the corresponding video $v \in \mathcal{V}$, aiming to produce an answer $a$ that closely matches the ground truth. This process can be approximated as:

$$\hat{a} = \arg\max_{a \in \mathcal{A}} F_\theta(a|q, v), \tag{14}$$

| Methods | DiDeMo (Video-to-text) | | | | | LSMDC (Video-to-text) | | | | |
|---|---|---|---|---|---|---|---|---|---|---|
| | R@1↑ | R@5↑ | R@10↑ | MdR↓ | MnR↓ | R@1↑ | R@5↑ | R@10↑ | MdR↓ | MnR↓ |
| Clip4clip Luo et al. (2022) | 41.4 | 68.2 | 79.1 | 2.0 | 12.4 | 20.8 | 39.0 | 48.6 | 12.0 | 54.2 |
| EMCL-Net Jin et al. (2022) | 45.7 | 74.3 | 82.7 | 2.0 | 10.9 | 22.2 | 40.6 | 49.2 | 12.0 | - |
| DiffusionRet Jin et al. (2023c) | 46.2 | 74.3 | 82.2 | 2.0 | 10.7 | 23.0 | 43.5 | 51.5 | 9.0 | 40.2 |
| T-Mass Wang et al. (2024) | 49.1 | 76.4 | 85.9 | 2.0 | **8.0** | **26.0** | 48.4 | **57.5** | 6.0 | **37.8** |
| SKRR | **50.4** | **77.8** | **86.5** | **1.0** | 8.6 | 25.9 | **48.5** | 57.4 | **6.0** | 38.1 |

| Methods | ActivityNet (Video-to-text) | | | | | Charades (Video-to-text) | | | | |
|---|---|---|---|---|---|---|---|---|---|---|
| | R@1↑ | R@5↑ | R@10↑ | MdR↓ | MnR↓ | R@1↑ | R@5↑ | R@10↑ | MdR↓ | MnR↓ |
| Clip4clip Luo et al. (2022) | 41.4 | 73.7 | 85.3 | 2.0 | 6.7 | - | - | - | - | - |
| EMCL-Net Jin et al. (2022) | 42.7 | 74.0 | - | 2.0 | - | - | - | - | - | - |
| DiffusionRet Jin et al. (2023c) | 43.8 | 75.3 | 86.7 | 2.0 | 6.3 | - | - | - | - | - |
| T-Mass Wang et al. (2024) | - | - | - | - | - | 13.2 | 37.3 | 48.5 | 11.0 | 56.1 |
| SKRR | **44.5** | **77.3** | **87.4** | **1.0** | **5.4** | **17.5** | **40.6** | **49.9** | **10.0** | **45.7** |

Table 6: Video-to-text retrieval performance on the DiDeMo Anne Hendricks et al. (2017), LSMDC Rohrbach et al. (2015), ActivityNet Krishna et al. (2017) and Charades Sigurdsson et al. (2016).

| Methods | MSRVTT>MSRVTT | | | MSRVTT>DiDeMo | | | MSRVTT>LSMDC | | |
|---|---|---|---|---|---|---|---|---|---|
| | R@1↑ | R@Sum↑ | MdR↓ | R@1↑ | R@Sum↑ | MdR↓ | R@1↑ | R@Sum↑ | MdR↓ |
| CLIP4Clip Luo et al. (2022) | 43.8 | 195.8 | 2.0 | 31.8 | 154.9 | 4.0 | 15.3 | 87.1 | 21.0 |
| X-Pool Gorti et al. (2022) | 46.9 | 201.9 | 2.0 | 35.3 | 168.5 | 3.0 | 16.4 | 93.5 | 18.0 |
| DiffusionRet Jin et al. (2023c) | 49.0 | 206.9 | 2.0 | 33.2 | 160.9 | 3.0 | 17.1 | 90.5 | 21.0 |
| T-Mass Wang et al. (2024) | 50.2 | 210.6 | 1.0 | 39.5 | 178.2 | 2.0 | 19.7 | 102.5 | 14.0 |
| SKRR | **51.0** | **211.0** | **1.0** | **41.2** | **183.6** | **2.0** | **20.2** | **105.0** | **12.0** |

Table 7: Text-to-video cross-domain generalization performance. $X > Y$, where $X$ denotes the training data and $Y$ denotes the testing data. Here, R@Sum=R@1+R@5+R@10.

where $\theta$ represents the trainable parameters group, $F$ represents the modeling function, and $q \in \mathcal{Q}$ and $v \in \mathcal{V}$. Following previous VQA methods Piergiovanni et al. (2022); Li et al. (2023), we directly apply DPSP$_{SP}$, DVFI$_{MaV}$, and DVFI$_{MiV}$ to process $v$ for the video question answering task, with the accuracy performance comparison presented in Table 8.

| Methods | Accuracy(%)↑ |
|---|---|
| Co-Tokenization Piergiovanni et al. (2022) | 45.7 |
| EMCL-QA Jin et al. (2022) | 45.8 |
| HBI Jin et al. (2023a) | 46.2 |
| TG-VQA Li et al. (2023) | 46.3 |
| SKRR | **47.2** |

Table 8: Video question answering performance on the MSRVTT-QA Xu et al. (2016).

## A.3 ABLATION STUDY

**Ablation Study of Sampling Frame Numbers.** In Table 9, we present a comparison of text-to-video retrieval performance under different numbers of sampled frames on the MSRVTT and Charades datasets. Since most MSRVTT videos are around 12 seconds long, increasing $N_f$ brings limited performance gains. For a fair comparison with existing methods, we uniformly set $N_f = 12$ on the MSRVTT dataset.

**Ablation Study of Feature Aggregation.** Feature aggregation refers to pooling multiple features into a single representation to enable direct similarity computation between text and video. In Figure 3, we present three mainstream feature aggregation methods and the similarity computation between single-instance and multi-instance features, including (a) Average, (b) MLP, and (c) MHA. In Table

| $N_f$ | MSRVTT (Text-to-Video) | | | | | Charades (Text-to-Video) | | | | |
|---|---|---|---|---|---|---|---|---|---|---|
| | R@1↑ | R@5↑ | R@10↑ | MdR↓ | MnR↓ | R@1↑ | R@5↑ | R@10↑ | MdR↓ | MnR↓ |
| 4 | 43.2 | 65.7 | 74.6 | 5.0 | 17.9 | 10.2 | 34.7 | 42.9 | 17.0 | 69.1 |
| 8 | 47.8 | 74.2 | 83.1 | 3.0 | 15.6 | 16.8 | 38.5 | 48.6 | 10.0 | 54.6 |
| 12 | 50.5 | 76.1 | 86.2 | 1.0 | 11.3 | 19.3 | 42.2 | 53.5 | 8.0 | 49.7 |
| 16 | 51.0 | 76.7 | 86.7 | 1.0 | 11.0 | 19.9 | 43.6 | 55.7 | 7.0 | 47.3 |
| 20 | **51.5** | 76.8 | 86.9 | 1.0 | 10.7 | **21.3** | 45.7 | 56.3 | 6.0 | 43.6 |

Table 9: Ablation study of different sampling frame numbers on the MSRVTT and Charades.

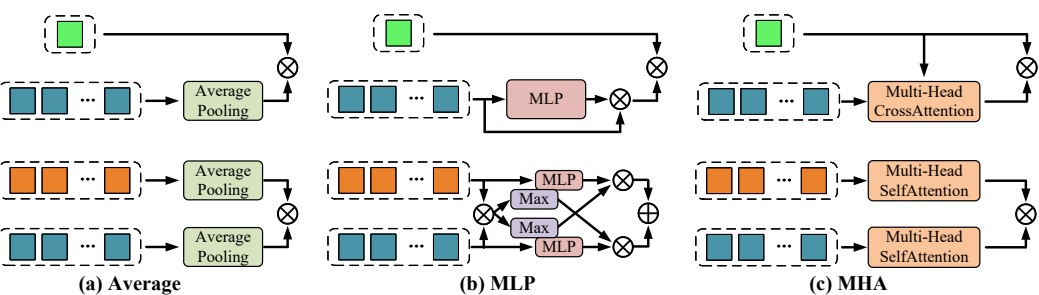

(a) Average  (b) MLP  (c) MHA

Figure 3: Feature Aggregation. The top part illustrates the similarity computation between single-instance and multi-instance features, such as $S_{T_s, V_f}$. The bottom part illustrates the similarity computation between multi-instance features, such as $S_{T_w, V_p}$. (a) Average: average pooling; (b) MLP: multilayer perceptron, used for linear mapping of feature weights; (c) MHA: multi-head attention mechanism, including both cross-attention and self-attention.

10, we report the retrieval performance of SKRR combined with the three feature aggregation methods. Note that MHA$_{\text{Average}}$ and MHA$_{\text{MLP}}$ indicate that the features are first fed into MHA, followed by feature aggregation using (a) Average or (b) MLP, respectively.

| Methods | R@1↑ | R@5↑ | R@10↑ | MdR↓ | MnR↓ |
|---|---|---|---|---|---|
| Average | 45.4 | 72.9 | 83.4 | 2.0 | 14.4 |
| MLP | 50.5 | 76.1 | 86.2 | 1.0 | 11.3 |
| MHA$_{\text{Average}}$ | 49.5 | 76.8 | 86.1 | 1.0 | 11.6 |
| MHA$_{\text{MLP}}$ | **51.4** | 77.3 | 87.5 | 1.0 | 10.4 |

Table 10: Ablation study of different feature aggregation on the MSRVTT.

**Ablation Study of KL Divergence.** Data leakage includes the following two common cases: Case **1**: the model has prior knowledge of the text paired with a video and uses this matched text to enhance the video. Case **2**: using VLMs to generate textual descriptions of videos, which are then directly used as query texts for retrieval. In Equation 12, we adopt knowledge distillation to jointly optimize the learning of the original text-video similarity. The main purpose of this process is to prevent retrieval data leakage and to simplify the sampling procedure. In Table 11, we compare the performance differences with and without using KL divergence. Compared with *w/o* KL, the *w* KL setting achieves superior performance (**+4.3%**).

| Methods | R@1↑ | R@5↑ | R@10↑ | MdR↓ | MnR↓ |
|---|---|---|---|---|---|
| *w/o* KL | 48.4 | 75.5 | 84.2 | 2.0 | 13.2 |
| *w/* KL | **50.5** | 76.1 | 86.2 | 1.0 | 11.3 |

Table 11: Ablation study of KL Divergence on the MSRVTT.

### A.4 VISUALIZATION

**Visualization of Recover-Path Benefits.** In Figure 4, we visualize the retrieval performance improvements achieved by incorporating the Recover-Path. On the MSRVTT test set, which contains 1,000 query texts and 1,000 candidate videos, the model constructs a $1000 \times 1000$ similarity matrix, where the 1,000 values along the main diagonal represent the similarities of positive pairs, and the off-diagonal values correspond to negative pairs. The results show that the similarity of positive pairs shifts to the right, while that of negative pairs shifts to the left, indicating that the Recover-Path effectively enhances the discriminability of correct matches.

**Visualization of Patch Compression.** In Figure 5, we illustrate the five-stage compression process of visual feature patches. Specifically, we select six frames from Video1409 and Video1308, and set the compression ratio for each step as $\lambda = 0.5$. We clearly observe that as the number of steps increases, the number of patch features gradually decreases, while the representation increasingly focuses on important entity features. This observation is consistent with human cognition of microscopic alignment.

**Visualization of Spot-Path Process.** In Figure 6, we visualize the frame selection and patch compression process within the Spot-Path during training. Red $\times$ marks indicate the discarded video frames, while the retained frames are used for patch feature compression. The last three rows show the visual results of the three successive rounds of patch compression.

**Visualization of Retrieval Results.** In Figure 7, we present a set of successful and failed retrieval cases during testing. Notably, most failures are attributed to the simplicity of textual descriptions and the high similarity among video scenes.

**Visualization of VQA Process.** In Figure 8, we visualize two examples of video question answering. The top part of the figure shows the query questions and corresponding video frames, while the bottom part presents the outputs from SKRR as well as the ground-truth answers. Both question-answering examples achieve 100% accuracy.

### A.5 ETHICS STATEMENT

This study does not involve human participants or the collection of personally identifiable information. The datasets employed in this research are publicly available and have undergone prior ethical review for academic use. We ensured compliance with relevant data licensing agreements and conducted all experiments in accordance with the terms of use of the datasets. No conflicts of interest or external sponsorship influenced the outcomes of this research.

Large language models (*e.g.*, GPT-4) were used solely as auxiliary tools to enhance the clarity and fluency of English writing. They did not contribute to the conception of ideas, the scientific content, data analysis, or the conclusions of this paper. We consider the use of LLMs in this limited capacity to pose no ethical concerns.

We have made every effort to ensure that our results are fully reproducible. The main paper and its appendix provide detailed descriptions of the proposed framework, including data processing pipelines, model architecture, and training procedures. For reproducibility, all datasets used in our experiments are publicly available, and we provide comprehensive descriptions of the preprocessing steps. We will release our code, data, and pretrained models.

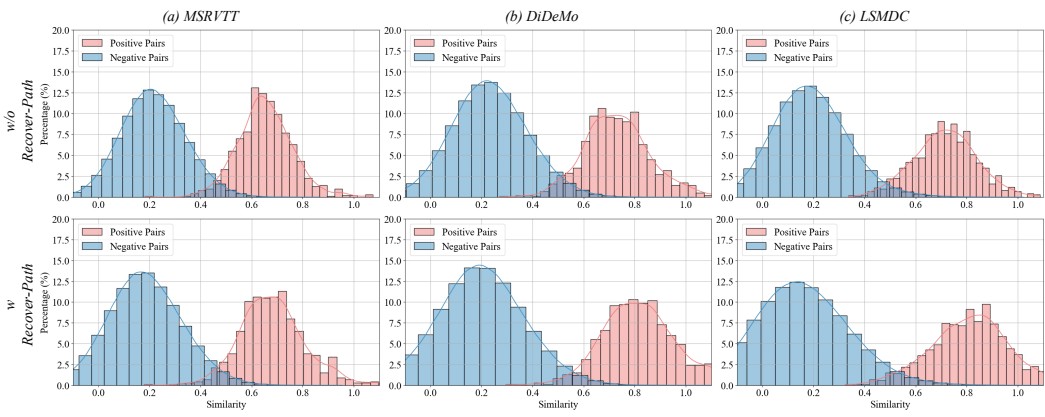

Figure 4: Visualization of similarity distributions under *w/o* Recover-Path and *w/* Recover-Path on the MSRVTT, DiDeMo, and LSMDC datasets. The x-axis represents similarity scores, and the y-axis indicates the proportion. From top to bottom, the similarity of positive pairs gradually increases, while that of negative pairs gradually decreases.

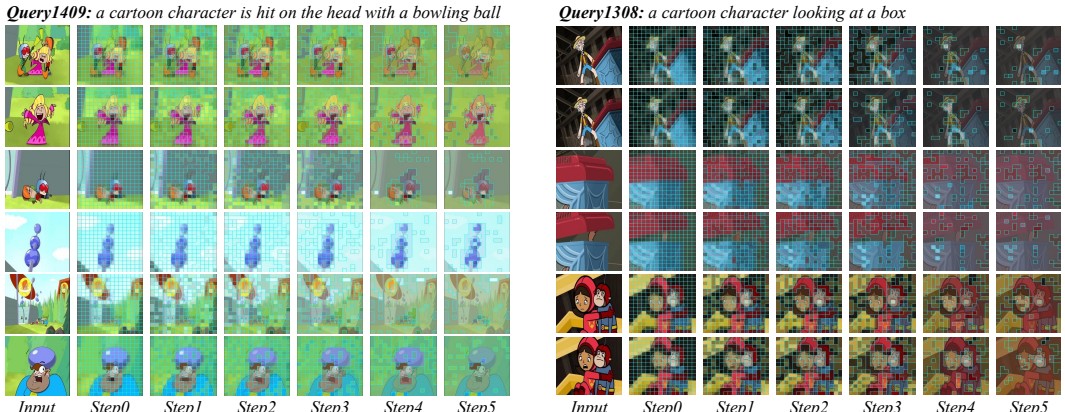

Figure 5: Visualization of the patch feature compression process. Six frames are selected from Video1409 and Video1308 on the MSRVTT, where the number of patches is reduced by half at each step. Note that a larger number of steps does not necessarily lead to better results; intuitively, Step 4 is sufficient to achieve the desired outcome.

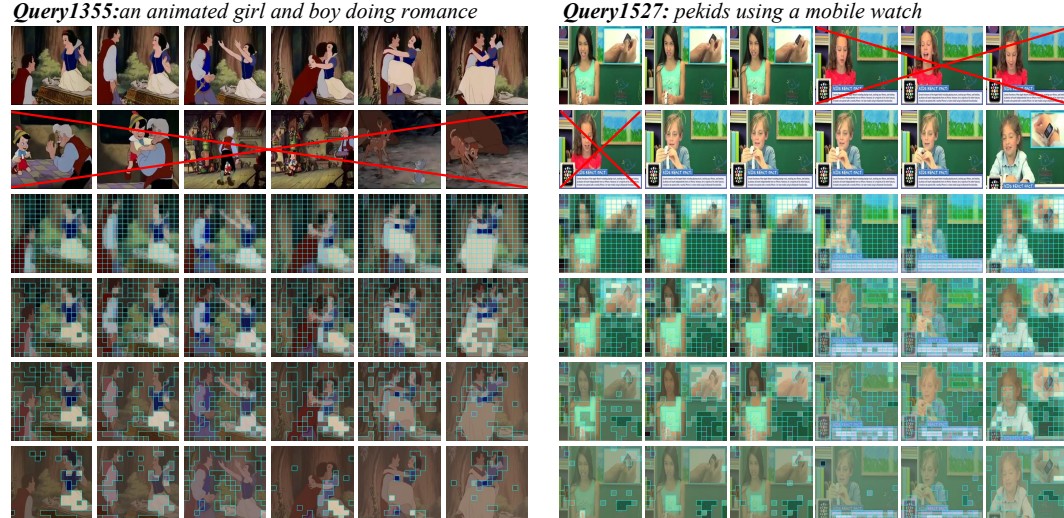

Figure 6: Visualization of frame selection and patch compression process within the Spot-Path during training. Red × indicates Recover-Path process similar to Spot-Path.

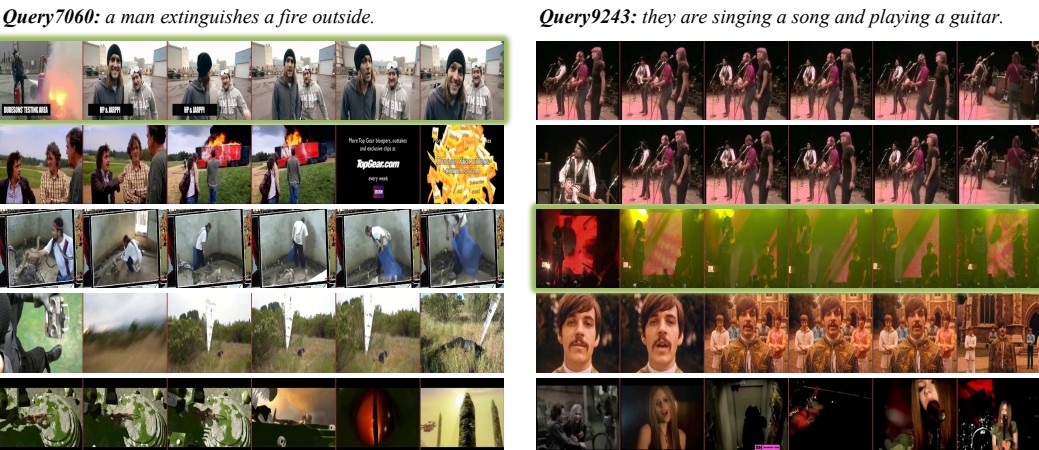

Figure 7: Visualization of successful case Query7060 and failed case Query9243. The retrieval results are displayed in descending order of similarity, where videos with green bounding boxes indicate the correct matches.

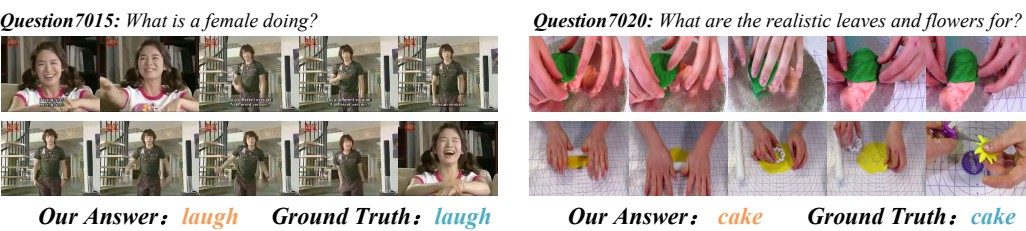

Figure 8: Visualization of video question answering. "Our Answer" denotes SKRR_QA's selected result from the candidate answers, while "Ground Truth" indicates the correct answer.

