# OpenReview forum: "Spot the Key, Recover the Rest: Dual-Path&View Representation Learning for Text-Video Retrieval"
_ICLR.cc/2026/Conference — ICLR 2026 Conference Withdrawn Submission_

### Official Review · Reviewer_pH13 · 2025-10-27

**Soundness:** 3
**Presentation:** 3
**Contribution:** 2
**Rating:** 4
**Confidence:** 4

**Summary:**

This paper proposes SKRR (Spot the Key, Recover the Rest), a CLIP-based text-video retrieval framework designed to enhance both feature representation and cross-modal interaction. The method introduces two key components: (1) the Dual-Path Feature Partitioning (DPFP) module, which separates visual features into highly relevant (Spot-Path) and complementary (Recover-Path) subsets; and (2) the Dual-View Feature Interaction (DVFI) module, which models alignment at both macro (sentence–frame) and micro (word–patch) levels. SKRR aims to simulate human cognitive processes in selecting and aligning visual-textual cues and achieves competitive performance across five benchmark datasets.

**Strengths:**

- **Well-structured dual-module framework:** The paper presents a clearly organized framework combining Dual-Path feature partitioning and Dual-View feature interaction, which together provide a coherent way to address both global and local cross-modal alignment. The conceptual motivation, inspired by human perception, is reasonable.

- **Complementary feature enhancement via recover path:** The introduction of the Recover-Path is a meaningful addition that leverages large vision-language models (e.g., GPT-4) to generate supplementary textual descriptions for low-similarity frames, effectively mitigating information loss in sparse-text scenarios.

- **Empirical Evaluation:** The authors conduct experiments on five standard benchmarks (MSRVTT, DiDeMo, LSMDC, ActivityNet, and Charades), demonstrating that SKRR consistently performs robustly across both short- and long-video retrieval settings.

**Weaknesses:**

- **Lack of Hyperparameter Analysis in clustering module:** The paper lacks analysis of key hyperparameters in the proposed clustering module. Specifically, the neighbor number k in the DPC-KNN algorithm (Eq. 6) is not specified, nor is its influence on clustering behavior or retrieval performance examined. Likewise, the iteration depth (number of compression steps) in the Patch Feature Compression Module (PFCM) is only visualized in Figure 5 but never quantitatively evaluated. Without ablation or sensitivity studies on these parameters, it remains unclear how robust the method is to different clustering settings or whether the reported performance depends on carefully tuned but unreported choices.

- **Lack of Key Comparative Experiments:** The experimental section lacks fair and comprehensive comparisons with several strong baselines from 2022–2023, such as CLIP-VIP (Xue et al., 2022) and its successors, whose reported R@1 scores (e.g.,55.9% on MSR-VTT with CLIP-ViT-B/32 and up to 57.7% with CLIP-ViT-B/16) substantially surpass those of SKRR (53.2% and 50.5% , respectively). These methods already achieve superior retrieval performance under the same CLIP backbone. Without direct quantitative comparison or discussion, it is difficult to justify the claimed state-of-the-art results. The absence of these key baselines significantly weakens the empirical validity of the paper’s contribution claims.

- **Lack of Novelty:** The overall methodological contribution of SKRR is limited. Both its Dual-Path (Spot/Recover) and Dual-View (Macro/Micro) designs largely follow established paradigms in prior text-video retrieval works, such as UCoFiA (Wang et al., ICCV 2023) and X-Pool (Gorti et al., CVPR 2022). The paper should explicitly clarify and analyze the key differences between SKRR and these prior works to better justify its claimed novelty.

**Questions:**

See weakness.

---

### Official Review · Reviewer_j2pq · 2025-10-27

**Soundness:** 3
**Presentation:** 2
**Contribution:** 2
**Rating:** 2
**Confidence:** 4

**Summary:**

The paper proposes SKRR, a text–video retrieval framework that first partitions visual features into highly relevant frames and low-relevance frames, which are textually completed by a VLM(GPT4), and then performs Dual-View Feature Interaction (DVFI) at macro and micro levels with a patch-compression module to curb redundancy. Experiments across five benchmarks show competitive performance.

**Strengths:**

1. The “two paths + two views ” story is intuitive and consistently executed; it mirrors human focus from global to local and avoids indiscriminate alignment that mixes signal and noise.

2. Without considering the model size, SKRR has achieved state-of-the-art (SOTA) performance compared to other non-general models.

**Weaknesses:**

1. The proposed "from key to the rest" and "from coarse to fine" strategies are common representation learning practices, with no clear evidence of distinctive novelty.  Additionally, CLIP-based methods (the framework’s foundation) are no longer a dominant trend—most state-of-the-art approaches now adopt VLM-based architectures. This misalignment with current research directions reduces the work’s significance and timeliness.


2. Although the model utilizes GPT-4 to provide rich text descriptions (which helps the model to realize text-video alignment better), the superiority of this model compared with previous methods is not significant. For example, the improvement over T-Mass on MSRVTT (ViT-B/32) is +0.3 R@1 (50.2→50.5), which—while non-trivial—feels small relative to the added system complexity.

3. Furthermore, the paper appears to lack sufficient comparison with the latest works in the field. Among the only two 2025 works mentioned—DiscoVLA and BiHSSP—their focus lies on parameter efficiency. If performance is to be compared with these two methods, the model’s computational cost should also be considered; however, this is not discussed.

4.In InternVideo2(proposed in 2024), even their 1B model achieves better performance than the proposed method. Notably, the proposed method employs GPT-4 as a component, resulting in no advantage in terms of computational overhead.

**Questions:**

1. Why are "from key to the rest" and "from coarse to fine" strategies critical to text-video retrieval tasks? It would be advisable to provide a summary of possible issues instead of a single negative sample(e.g., if one strategy is absent, what kind of problems will happen?)

2. Why are previous methods unable to implement the "from key to the rest" and "from coarse to fine" strategies? What are the difficulties or concerns in implementing these two strategies for them? I think providing such a discussion can help enhance the significance of this paper.

3. Can you discuss the efficiency of your model? It's necessary to do so if you want to compare with DiscoVLA and BiHSSP. It may also help explain why VLMs are absent in the comparison experiment if you have many fewer parameters.

---

### Official Review · Reviewer_pAVQ · 2025-10-31

**Soundness:** 2
**Presentation:** 3
**Contribution:** 2
**Rating:** 2
**Confidence:** 5

**Summary:**

This paper proposes SKRR, a CLIP-based text-video retrieval framework. It introduces a dual-path feature partitioning (DPFP) module to enhance features by separating and textually supplementing less relevant visual content, and a dual-view feature Interaction module for coarse-to-fine granularity alignment. The method synergistically improves cross-modal interaction, achieving state-of-the-art performance on five benchmark datasets, including 50.5% R@1 on MSRVTT.

**Strengths:**

The strengths of this paper lie in the following aspects:
1) This paper has clear modular design. That is dual-path feature partitioning for enhancement and dual-view feature interaction for alignment, with explicit, formula-backed pipelines. The motivation behind these designs are interesting and inspiring.
2) The ablation seems effective, Systematic toggles of Spot/Recover and Macro/Micro (Table 3), granularity configurations (Table 4), and selection/compression factors (Table 5) demonstrate where gains arise (notably +7–8 R@1 over baseline). Sensible operating point at (ℏ, λ̄) = (0.50, 0.50) with explicit runtime/memory trade-offs (Table 5).
3) As well, the empirical performance also shows the performance. noteworthy gains on long-video (DiDeMo R@1 +14.3% vs X-Pool) and sparse-text (LSMDC R@1 +14.5% vs CLIP-VIP) settings.
4) The presentation of this paper is good and very straightforward to follow. I donot doubt the reproductivity of this proposed approach.

**Weaknesses:**

The major concerns towards this paper is: In this paper, VLMs, e.g., GPT-4 is adopted to generate textual descriptions for lowsimilarity frames as complementary interaction targets. That means the improved gains highly depend on a external strong VLMs, and this is un-fair for other approaches with this external tool. How sensitive is the final gains to the ability of the VLMs? If the ability of the VLMs is weakened, how about the performance?
Use of external VLM/LLM augmentation is compared to some expansion baselines, but prompts, model versions, number/length of generated captions (N_s^=), and filtering thresholds are insufficiently specified.

Some other technical question:
1) TopN-based frame/text selection (Eqs. 3–4) introduces discrete sampling without discussion of gradient flow or stability; potential training sensitivity.
2) Despite patch compression, end-to-end GPU memory and test time increase vs baseline (Table 3), and DPC-KNN adds O(Np^2) neighbor computations per iteration; no complexity analysis or k sensitivity reported.
3) There are possible ambiguity at inference. It is unclear whether VLM-generated texts are used only during training; final similarity aggregation lacks weighting/normalization details.
4) No analysis of sensitivity to noisy or adversarial queries; selection may amplify errors when sentence–frame similarity is unreliable.
5) Heuristic choices (density metric, \delta rule, attention bias I) lack ablations against simpler baselines (e.g., top-k patch scoring) or theoretical justification.
6) While authors mention avoiding “data leakage", selecting frames using the query and augmenting with video-derived captions may inadvertently create overly aligned pairs; clearer safeguards and protocol details are needed.

**Questions:**

Please see the weakness section.

---

### Official Review · Reviewer_ELw4 · 2025-11-01

**Soundness:** 2
**Presentation:** 2
**Contribution:** 2
**Rating:** 2
**Confidence:** 4

**Summary:**

The paper proposes a CLIP-based text-video retrieval framework named SKRR, which introduces two complementary modules—the Dual-Path Feature Partitioning (DPFP) for feature enhancement and the Dual-View Feature Interaction (DVFI) for multi-granularity alignment. Experimental results demonstrate the proposed method can achieve good performance.

**Strengths:**

1) The paper is well written and clearly structured, with a coherent motivation and intuitive framework design.
2) Experimental results demonstrate strong performance, achieving high retrieval accuracy.

**Weaknesses:**

1) Although the proposed model achieves excellent retrieval performance, it likely comes at the cost of increased computational complexity. The paper should report and analyze the training and inference costs (e.g., FLOPs, GPU hours, latency).

2) In Eq. (4), the number of generated augmented texts is not clearly defined or experimentally explored. A detailed discussion on how this hyperparameter affects model performance and stability can make the methodology more complete.

3) The use of an MLP for feature aggregation is standard and has already been widely adopted in prior work such as Cap4Video. Moreover, the design of the PFCM module for Micro Feature Interaction appears to be largely derived from HBI, limiting the methodological originality.

4) In the inference stage, through the calculated all similarity matrices S_{Ts}, V_f, S_{Ts}, V_p and S_{Tw}, V_p
and then aggregated into a final similarity matrix, there is no clear explanation of the selected aggregation mode (average or weighted).

5) The baseline methods used for comparison appear to be inconsistent across different datasets (e.g., MSRVTT, DiDeMo, and LSMDC).

6)  The proposed approach appears to have limited innovation, as it mainly combines or extends existing components rather than introducing a fundamentally new idea.

**Questions:**

See weaknesses section.

---

### Note · Authors · 2026-01-23

I have read and agree with the venue's withdrawal policy on behalf of myself and my co-authors.